# Advances in OCT Imaging in Myopia and Pathologic Myopia

**DOI:** 10.3390/diagnostics12061418

**Published:** 2022-06-08

**Authors:** Yong Li, Feihui Zheng, Li Lian Foo, Qiu Ying Wong, Daniel Ting, Quan V. Hoang, Rachel Chong, Marcus Ang, Chee Wai Wong

**Affiliations:** 1Singapore National Eye Centre, Singapore Eye Research Institute, Singapore 169856, Singapore; liyong@u.duke.nus.edu (Y.L.); zheng.feihui@seri.com.sg (F.Z.); foo.li.lian@singhealth.com.sg (L.L.F.); wong.qiu.ying@seri.com.sg (Q.Y.W.); daniel.ting@duke-nus.edu.sg (D.T.); gmshqvm@nus.edu.sg (Q.V.H.); rachel.chong.s.j@snec.com.sg (R.C.); marcus.ang@singhealth.com.sg (M.A.); 2Ophthalmology and Visual Sciences Academic Clinical Program, Duke-NUS Medical School, Singapore 169857, Singapore; 3Department of Ophthalmology, Yong Loo Lin School of Medicine, National University of Singapore, Singapore 119077, Singapore; 4Department of Ophthalmology, Columbia University, New York, NY 10027, USA

**Keywords:** optical coherence tomography (OCT), optical coherence tomography angiography (OCTA), myopia, pathologic myopia, imaging

## Abstract

Advances in imaging with optical coherence tomography (OCT) and optical coherence tomography angiography (OCTA) technology, including the development of swept source OCT/OCTA, widefield or ultra-widefield systems, have greatly improved the understanding, diagnosis, and treatment of myopia and myopia-related complications. Anterior segment OCT is useful for imaging the anterior segment of myopes, providing the basis for implantable collamer lens optimization, or detecting intraocular lens decentration in high myopic patients. OCT has enhanced imaging of vitreous properties, and measurement of choroidal thickness in myopic eyes. Widefield OCT systems have greatly improved the visualization of peripheral retinal lesions and have enabled the evaluation of wide staphyloma and ocular curvature. Based on OCT imaging, a new classification system and guidelines for the management of myopic traction maculopathy have been proposed; different dome-shaped macula morphologies have been described; and myopia-related abnormalities in the optic nerve and peripapillary region have been demonstrated. OCTA can quantitatively evaluate the retinal microvasculature and choriocapillaris, which is useful for the early detection of myopic choroidal neovascularization and the evaluation of anti-vascular endothelial growth factor therapy in these patients. In addition, the application of artificial intelligence in OCT/OCTA imaging in myopia has achieved promising results.

## 1. Introduction

Myopia represents a growing and significantly global public health problem, with a prevalence of over two billion people (28.3% of the global population), including 277 million individuals (4.0%) with high myopia [1]. High myopia is defined as a refractive error of −5.00 diopters (D) or worse [2]. Pathologic myopia is defined by the presence of myopic lesions in the posterior segment of the eye (posterior staphyloma or myopic maculopathy equal to or more serious than diffuse choroidal atrophy) [2]. It usually occurs in highly myopic eyes, but can also develop in eyes with low myopia or even in emmetropia. It has been reported that pathologic myopia causes vision impairment or blindness in 0.2–1.5% of the Asian population and is the leading cause of irreversible blindness in China, Japan, and Taiwan [3,4,5,6]. Therefore, myopia and pathologic myopia have become serious threats to vision health worldwide warranting prompt attention [7].

Imaging in myopia has become increasingly important for the early detection, accurate diagnosis, prognostication, and evaluation of treatment for myopia [8]. Optical coherence tomography (OCT), an emerging technology for performing high-resolution cross-sectional imaging, is now the most frequently used tool for ophthalmic imaging. Before the advent of OCT, the pathological changes of the myopic eyes could only be histologically investigated in enucleated eyes. Now, OCT can be applied for imaging characteristic changes in ocular tissue due to myopia [9], from the anterior to posterior segment of the eye, including cornea, anterior chamber, vitreous, retina, optic nerve, choroid and sclera, which has greatly improved our understanding of myopia and pathologic myopia.

More recently, the development of OCT angiography (OCTA) has enabled the production of images of blood flow with unprecedented resolution of all the vascular layers in a rapid, non-invasive manner, which can help detect vascular-related myopia-related macular lesions such as myopic choroidal neovascularization (mCNV) and myopia-associated glaucoma-like optic neuropathy [10,11]. On the other hand, artificial intelligence (AI) has been increasingly applied in the diagnostic and prognostication of ophthalmic diseases, including myopia and pathologic myopia [12,13]. In particular, deep learning algorithms driven by OCT images have proven powerful in detecting macular and optic nerve complications related to myopia [14]. 

This review summarizes recent findings in the advancement of OCT/OCTA in imaging myopia and pathologic myopia from the anterior to posterior segment, and discusses the clinical implications of these findings, especially in combination with AI. We also discuss the potential challenges and future research directions of OCT/OCTA in myopia.

## 2. Advances in OCT/OCTA Technology

### 2.1. Time Domain and Spectral Domain OCT

The invention of OCT in 1991 [15] has significantly impacted the entire field of ophthalmology by enabling real-time in vivo subsurface imaging of biological tissue in a non-invasive manner. Since the advent of OCT, there have been extraordinary advances in this imaging technology. The first generation of time-domain OCT (TD-OCT) was limited by low axial resolution (10–15 μm) and longer time for imaging due to limited number of A-scans (400 A-scans/second) [16]. The second generation of spectral domain OCT (SD-OCT) improved axial resolution (3–5 μm) and image acquisition speed (20,000–100,000 A-scans/second) [17]. SD-OCT enables the visualization of retinal microstructures to the same level of detail as histopathology [18]. By changing the imaging acquisition position, enhanced depth imaging (EDI, which allows imaging of deeper structures such as the choroid and sclera [19]) has been incorporated in commercially available SD-OCT. OCT has become a standard of care in ophthalmology for disease diagnosis, monitoring of progression and treatment response, and advancing our understanding of disease pathogenesis. 

### 2.2. Swept Source OCT

Swept source OCT (SS-OCT) uses a frequency-swept light source and photodetector. The light source is intrinsically more complex and the detectors are able to operate at high speed (200,000 A-scans/second), which allows faster image acquisition and reduces motion artifacts. In addition, SS-OCT employs lasers with a longer wavelength, which can better penetrate through ocular tissues with less sensitivity roll-off, thus increasing imaging depth and improving visualization through dense ocular media, though at the expense of reduced axial resolution (6–8 μm) [20]. Studies have demonstrated that SS-OCT is helpful for imaging the retinochoroidal structures of pathologic myopia and revealing additional pathology along the staphyloma walls not visible using SD-OCT, including incomplete posterior vitreous detachment (PVD) and peripheral retinoschisis [21].

### 2.3. OCT Angiography

Established on the basic principles of OCT, OCTA can achieve non-invasive, depth-resolved, and refined imaging of the choroid and retina microstructures. The concept of imaging vasculature through OCT and Doppler shift formed the basis for OCTA as OCT scanning speeds improved dramatically [22]. OCTA generates images through calculating differences in phase, amplitude, or both between sequential OCT scans at the same spot, known as a decorrelation signal, which is yielded by moving architecture. For example, the movement of blood cells within retinal blood vessels yield a decorrelation signal, allowing OCTA images to highlight retinal microvasculature [23]. In addition, quantitative OCTA analysis of chorioretinal microvasculature, such as blood vessel density, tortuosity, and caliber, etc., have been established for objective description and explanation of clinical outcomes [24].

### 2.4. Widefield OCT

The introduction of widefield and ultra-widefield phenotypes represent one of the most important recent advances of OCT/OCTA. Ultra-widefield SS-OCT has been reported to scan a field of view up to 100° of the retina [25], thus enabling the detection of wide staphyloma as well as estimation of ocular curvature [26,27]. Ultra-widefield OCT is extremely useful for imaging peripheral retinal lesions in high myopia, some of which had not been previously imaged. The advent of widefield and ultra-widefield OCTA has greatly expanded the field of view of 50–80° of the retina and have been employed for the detection and treatment of various chorioretinal diseases [23,28].

### 2.5. Polarization Sensitive OCT

Polarization sensitive OCT (PS-OCT) bears added advantages owing to the fact that several ocular tissues can change the polarization state of light, creating an extra contrast channel and providing quantitative information, which has shown to be useful for both anterior and posterior segment imaging, including the evaluation of trabecular meshwork, retinal nerve fiber layer (RNFL) and macular lesions [29]. PS-OCT has shown potential utility in evaluating myopia-related pathologic changes. For example, PS-OCT successfully demonstrated the structure of collagen fibers of the retinal nerve fibers, sclera, and Henle’s fiber layer [30]. In addition, PS-OCT was used to assess melanin distribution at the retinal pigment epithelium (RPE) in high myopia patients and showed decreased depolarization at the RPE [31].

## 3. OCT and OCTA for the Assessment of Ocular Structures in Myopes

### 3.1. Anterior Segment

Anterior segment OCT (AS-OCT) imaging is increasingly influencing clinical practice. AS-OCT can be used to assess modest anterior segment morphological changes, such as slight reductions in the accommodation response of ocular biometric elements elicited by atropine eye drops, which are used for childhood myopia control [32]. AS-OCT is also useful in the pre-operative, intra-operative, and post-operative evaluations of corneal refractive surgery candidates [33]. Ultra-high-resolution OCT can effectively detect subclinical keratoconus [34,35], an important part of pre-operative screening for refractive surgery. High-resolution AS-OCT is useful for the measurements of corneal thickness, corneal keratometry, flap thickness or displacements after laser in situ keratomileusis (LASIK) [36,37]. The curvatures of the stromal layers can also be measured to visualize the reconstruction of the stroma and epithelium after photorefractive keratectomy (PRK) [38]. For example, using ultra-high-resolution AS-OCT, corneal epithelial hyperplasia after PRK was found to be associated with preoperative myopic error and ablation zone diameter [39]. In addition, AS-OCT has an important role in determining the pre-operative ocular biometrics of implantable collamer lenses (ICL) implantation for myopes [40], which provides reliable basis for the optimization of ICL sizing [41]. The important safety postoperative parameter known as the lens vault, which is the distance between the anterior surface of the crystalline lens and the posterior surface of the ICL [42], can also be measured using AS-OCT [43]. Moreover, AS-OCT has been used to investigate the development and risk factors for intraocular lens (IOL) tilt and decentration in high myopes [44]. Extra precautionary measures should be taken before implanting multifocal or toric IOL in such eyes.

Recently, studies have shown that anterior segment OCTA is able to non-invasively measure functional vascular parameters that may assist the evaluation of ocular surface diseases [45]. Future studies are necessary to explore its potential applications in myopia-related diseases.

### 3.2. Vitreous

OCT technology has significantly improved the imaging of vitreous structure with advancements in depth of view [46]. Recently, the combination of ultra-widefield OCT and 3D imaging has shown to be very useful in visualizing the complex structure of the posterior vitreous [47]. The posterior precortical vitreous pocket (PPVP) is defined as the liquefied lacuna anterior to the macular and is physiologically present in some adults [48]. Using SS-OCT, PPVPs have found to be larger in highly myopic eyes, which may indicate earlier vitreous liquefaction that may induce partial or complete PVD in these patients [49]. Recently, a higher incidence of asymmetrical PVD was found in high myopia, and correlated with posterior protrusion of the sclera of the highly myopic eye, which suggests that myopic macular retinoschisis and myopic traction maculopathy (MTM) may be induced by vitreous traction spanning a wider distance in these eyes [50]. In addition, SD-OCT has also been used to confirm PVD in myopic eyes, which is associated with structural and functional abnormalities such as vision-degrading myodesopsia [51].

### 3.3. Retina

High myopia and pathologic myopia are featured by excessive elongation of the eyeball that can lead to a series of retinal complications including retinoschisis, lacquer cracks, myopic maculopathy, and myopic macular hole (MH) [52]. OCT can be used to qualitatively and quantitatively evaluate the chorioretinal structures in high myopes [53]. Using SD-OCT, a correlation was found between the degree of myopia and the change of retinal thickness. With the increase in myopia, the foveal thickness increased while the inner/outer macular thickness decreased [54]. In addition, the morphology of Bruch’s membrane defects visualized by SS-OCT were consistent with prior histopathology studies in high myopia [55,56], which were characterized by a lack of Bruch’s membrane, choriocapillaris, photoreceptors, and RPE [56]. It is important to visualize and image the pathological features in the peripheral retina for the diagnosis and evaluation of pathology in the area [57]. Ultra-widefield OCT imaging of the retinal periphery is feasible with commercially-available devices which provide detailed anatomic information of the peripheral retina [58]. Advances in ultra-widefield OCT technology address many challenges and allow new findings of the structure and function in high myopes. For example, ultra-widefield OCT can visualize the staphylomatous contour of highly myopic eyes, generating detailed imaging of the vitreoretinal interface and progressive lesions of MTM [59].

Above all, it is important to consider the magnification correction for the accurate interpretation of OCTA-derived parameters in myopia, since ocular magnification can affect the outcomes of chorioretinal blood flow quantification with OCTA [60]. Retinal perfusion has been found to be associated with chorioretinal atrophy, mCNV, and lacquer crack formation in myopic eyes [61,62]. Studies using SD-OCTA have reported disparity in correlated changes to the deep versus superficial retinal circulations, indicating that retinal circulations can be affected by high myopia. SS-OCTA can be used to quantitatively assess the retinal microvasculature and choriocapillaris in myopes [63]. Compared with low and moderate myopia, vessel density of the superficial capillary plexus was lower, and impairment of the choriocapillaris in the macular area was more severe in the high myopia group [63]. Widefield SS-OCTA can generate detailed images of the chorioretinal microvasculature in a large field of view, with which researchers found that decreases in microvasculature and structural changes were correlated with myopia [64].

### 3.4. Choroid

The choroid plays an important role in the pathological alterations of myopia and myopia-related complications [65]. It has been demonstrated that the extreme thinning of the choroid can cause decreased choroidal perfusion, which may induce the development of mCNV and myopic macular degeneration (MMD) [66]. Using SD-OCT, choroidal thickness has been identified to be associated with axial length and visual outcomes in high myopes [67,68]. SS-OCT has the advantage of generating high-resolution images of the choroid and the choroid–scleral interface [69]. Using SS-OCT, researchers found that choroidal thinning actually developed before retinal thinning occurred during myopia progression; myopia shift was found to be independently correlated with central fovea choroidal thinning and axial length [70], indicating the value of SS-OCT to clarify the critical roles of choroid and retina during myopia progression. In addition, choroidal thinning was also found to be correlated to MMD [71], and topographic variations were identified in 3D maps of choroidal thickness [72]. Another important choroidal feature is peripapillary intrachoroidal cavitation, usually located inferior to the optic nerve in highly myopic eyes [73]. Contrary to the traditional hypothesis that intrachoroidal cavitation was an elevation of the retina and RPE, studies using EDI SD-OCT and SS-OCT reported that it could also develop sparing the alteration of the retina and RPE through posterior scleral bowing [73]. In addition, intrachoroidal cavitation was identified in 4.9% of highly myopic eyes, of which 71% were associated with glaucomatous visual field defects [74]. Nevertheless, the underlying pathogenesis of intrachoroidal cavitation development and its association with visual field defects remains unelucidated.

OCTA has also been used to explore the choriocapillaris in myopic eyes [63,75,76,77]. Researchers developed a novel segmentation technique based on OCTA to assess different choroidal layers’ thickness and vasculature, which may serve as new biomarkers to study myopia-related complications [76]. In addition, studies showed more severe choriocapillaris flow defects in high myopia, even in those with mild fundus changes, in both standard scans and quantitative mapping [63,75], which worsened with increasing severity of myopic maculopathy [77]. More specifically, quantitative OCTA measurements of the chorioretinal microvasculature in high myopes demonstrated that decreased retinal perfusion was related with axial length elongation [78]. Using SS-OCTA, studies showed that retinal perfusion and choriocapillaris flow decreased with worsening severity of MMD, indicating that both the retinal and choriocapillaris vasculature are affected in eyes with MMD [79]. More recently, studies using SS-OCTA to measure retinal perfusion density demonstrated that macular sensitivity was associated with deep retinal perfusion density, indicating a vasculature-function correlation in MMD development [80].

## 4. OCT and OCTA for the Assessment of Pathology in High Myopes

### 4.1. Myopic Maculopathy

Myopic maculopathy is defined as “macular alterations induced by high myopia, in which an excessive axial length and/or posterior staphyloma is the main common factor but not the only factor” [81]. The META-PM classification is based solely on fundus photographs showing only the atrophic alterations of myopic maculopathy [82]. However, macular alterations include atrophic changes in addition to neovascular alteration, traction-induced changes, and dome-shaped macula (DSM), which can only be visualized and diagnosed with OCT. Therefore, an OCT-based classification system has been proposed [83]. Using SS-OCT, the cut-off value of 56.5 μm for the nasal choroidal thickness can predict peripapillary diffuse choroidal atrophy from the tessellation, and the cut-off value of 62 μm at subfoveal can predict the macular diffuse choroidal atrophy. During progression from macular diffuse choroidal atrophy to patchy atrophy, other factors besides choroidal thinning, including defect of Bruch’s membrane may also be involved [83].

Recently, researchers have proposed another new classification system of myopic maculopathy based on the three key factors—Atrophy, Traction, and Neovascularization, which is known as the ATN classification system [82]. It keeps the original META-PM classification as the atrophy classification; adds the tractional component including five stages of inner or/and outer foveoschisis, foveal detachment, macular hole, and retinal detachment; three plus signs in the META-PM classification are included as neovascular components [82].

#### 4.1.1. Atrophic Myopic Maculopathy

Atrophic myopic maculopathy progression is featured by choroidal thinning, which can be seen on EDI SD-OCT, SS-OCT, and OCTA (Figure 1). Recent studies of OCT/OCTA imaging in MMD can be seen in Table 1, and have been discussed in Section 3.4. 

#### 4.1.2. Myopic Traction Maculopathy (MTM)

The development of OCT allows better visualization of detailed alterations and early insights of MTM, which can help not only the classification and definition of the disease, but also the assessment of the natural course [85]. Using widefield SS-OCT, researchers found that coexisting extrafoveal retinoschisis were frequently identified in the inferotemporal area and were related to retinal vessel, microfolds, microbreaks, and staphyloma [86]. Recently, a new classification system of MTM based on OCT images has been proposed: the retinal stages 1–4 are presented along the vertical axis emphasizing the evolution in the layers perpendicular to the retinal plane; the foveal stages a–c are presented along the horizontal axis emphasizing the evolution in the layers tangential to the retinal plane; the occurrence of an outer lamellar macular hole or epiretinal abnormalities is recorded as “O” or “+” [85]. According to the new grading system, indication for surgical intervention should be considered for stages 3 and 4, or any stage b and c, in order to preserve and improve vision in MTM patients [85]. The advice of surgical treatment with macular buckle or pars plana vitrectomy is listed for each different stage [87]. In addition, OCTA has also been used to describe the imaging characteristics of MTM and detect potential biomarkers, which may further elucidate the pathogenesis of MTM [88].

#### 4.1.3. Myopic Choroidal Neovascularization (mCNV)

As one of the most sight-threatening retinal complications of pathologic myopia, mCNV can cause an abrupt onset but gradual deterioration of central vision [89]. The diagnostic standard of mCNV is by imaging using fundus fluorescein angiography (FFA), which shows neovascularization as well-defined hyperfluorescence in the early stage and leaking in the later stage [2]. Studies demonstrated that different types of mCNV can be detected by FFA, and the specific findings were found to be consistent with SD-OCT scan results [90]. Recently, SD-OCT gradually surpassed FFA as the primary method for evaluating disease activity due to its non-invasive nature (Table 2) [91]. On SD-OCT, active mCNV presents as a dome-shaped hyperreflective subretinal lesion with ill-defined margins, usually with minimal subretinal fluid (Figure 2) [92]. The two-step diagnostic approach combining external limiting membrane distortion with RPE elevation on OCT showed excellent sensitivity and specificity for diagnosing mCNV [91]. After treatment, the hyperreflective lesion consolidates and acquires a distinct border [8].

OCTA is also very helpful for detecting mCNV noninvasively with high sensitivity and specificity (Table 2). SD-OCTA has shown to be useful for the early detection as well as treatment monitoring of mCNV (Figure 2) [93]. The fundamental benefit of OCTA is that it is non-invasive, allowing for multiple scans to be examined during each follow-up visit. However, generally OCTA cannot achieve accurate evaluation of the disease activity. Therefore, it is recommended to use multi-model imaging such as OCTA together with OCT to fully assess mCNV [8], which may show higher sensitivity than each modality alone [94]. Recently, however, a unique diagnostic technique using OCTA combining mCNV appearance, anastomoses/loops, and vascular branching has been proposed, which may be a useful tool for assessing the disease activity of mCNV [95]. Interestingly, researchers found an excellent agreement in mCNV area measurements and a poor agreement in mCNV vessel density measurements for SS-OCTA devices in automatic and manual segmentations [96]. In addition, OCTA is also useful to evaluate the anti-vascular endothelial growth factor (VEGF) therapy for mCNV [97,98,99], in which the vessel junctions may be one of the most important biomarkers [100]. OCTA-based analysis generated both intuitive images and quantitative data, which could lead to new perceptions of treatment response evaluation for mCNV [100]. However, OCTA bears several limitations. Due to a longer acquisition time, blinking or loss of fixation can lead to motion artifacts presented as dark bands in the image [101]. Projection artifacts can also occur, such as the projection of the superficial retinal vascular networks onto the choriocapillaris layer [101]. In addition, the performance of image segmentation can drop when the retinal structure is interrupted by mCNV.

### 4.2. Dome-Shaped Macula (DSM)

DSM has been better depicted and visualized with the advancement of OCT technology that has allowed acquisition of high-resolution images of the macula area [103]. According to the primary orientation of the macula based on OCT images, three different types of DSM have been recorded: round dome (20%), vertical oval-shaped dome (18%), and horizontal oval-shaped dome (62%) [103,104]. Studies using 3D OCT macular reconstruction and 3D MRI have confirmed these findings since the conventional OCT equipment has a restricted scan length and may not achieve comprehensive evaluation [105,106]. Using SS-OCT, one study demonstrated a correlation between Bruch’s membrane defects and the occurrence of DSM in high myopia [55]. In addition, using ultra-widefield SS-OCT, a recent study reported that DSMs develop independently from staphyloma, and tend to form in eyes with a substantial enlargement of the posterior fundus and should be regarded as scleral curvature deformation [26]. 

### 4.3. Optic Nerve

Several studies have reported the myopia-related complications in the optic nerve and peripapillary area, such as peripapillary atrophy (PPA) enlargement [107], tilted and rotated discs [108,109], scleral thinning, and abnormality between the macular and the optic nerve head (Figure 3) [110], which have been found to be accountable for the predisposition to glaucoma. Demonstrating the structural abnormalities in the optic nerve and surrounding area may aid in elucidating the underlying etiology of myopia-associated glaucoma-like optic neuropathy and detecting high-risk eyes of developing glaucoma.

It was reported that a higher incidence of glaucoma in axial high myopia is due to the disc enlargement in axial myopia, instead of the elongated axial length [111]. In most high myopes, an OCT scan provides enough information to diagnose glaucomatous optic neuropathy accurately [112]. SS-OCT can be a helpful method to visualize peripapillary morphologic characteristics in high myopia. The peripapillary area is frequently reported to be inferotemporal tilted in highly myopic eyes using SS-OCT [113]. Deeper staphyloma, thinner choroid, and larger area of PPA were found to be associated with greater tilt [113]. Recently, researchers published a public dataset of SD-OCT images of optic disc tilt in myopia, aiming to develop new findings between optic disc tilt and pathologic myopia [114]. Study using high-resolution SS-OCT suggested that eyes with sudden alterations to the sclera have greater visual field defects than those without, and the angle of scleral bending was correlated with the thickness of retinal nerve fiber layer and visual field defect in high myopes [110]. In addition, there was a correlation between tilted disc ratio measured by SS-OCT and retinal perfusion measured by SD-OCTA in high myopia [115]. Choroidal thinning of the macular area was correlated with optic disc tilt degree and increased PPA area, and macular choroidal microvasculature changes were also correlated with increased PPA area [116]. 

OCTA measurements of calculated indices such as average macular vessel density ratio have proved to be helpful for diagnosing glaucoma in high myopic eyes (Figure 3) [117]. Primary open-angle glaucoma eyes with high myopia had a higher rate of decrease in macular vessel density in the deep capillary plexuses than those without high myopia, which may provide new clues to the longitudinal effects of high myopia in retinal microvasculature [118]. Using OCTA, topographic differences on choroidal microvasculature and superficial radial peripapillary capillary were identified in PPA subzones, indicating there may be a microcirculatory deficiency in PPA beta zone in myopia [119]. Choroidal microvascular dropout observed by OCTA were found in highly myopic glaucoma eyes and were topographically associated with the region of visual field defects, which may facilitate diagnosis of glaucoma in high myopia [120]. In addition, a study found that the widefield SS-OCTA vascular density map showed good diagnostic ability for the detection of glaucomatous alterations in high myopes, which was stronger than traditional imaging methods such as OCT or red-free fundus photography [121].

### 4.4. Sclera and Posterior Staphyloma

The sclera plays an important role in the development of myopia and the related retinal complications through its biochemical and biomechanical properties. OCT can be helpful in the investigation of the sclera, revealing subtle abnormalities in posterior staphyloma, and allows identification of the spatial relationship between the protruded sclera and morphology of the retinal and choroidal layers [122]. In particular, SS-OCT is now able to generate high-resolution images of the entire sclera in one single scan with even more refined architectural study. 

The abnormalities found in the shape and thickness of the sclera in myopic eyes have led to some new perceptions into the pathophysiology of myopia. For example, it is reported that the choroid and sclera are thinner in myopic eyes than in normal eyes, and the subfoveal scleral thickness were reported to be associated with refractive error, choroidal and retinal thickness, and age [123]. Using SS-OCT and 3D MRI, researchers studied the morphology of the sclera in highly myopic eyes to understand the pathogenesis of myopic retinochoroidal lesions [122]. Irregular curvature of the inner sclera, which may cause abnormal stress of retinal RNFL and vitreomacular interface, was reported to be related to higher incidence of myopic maculopathy including MTM, mCNV, and chorioretinal atrophy [122]. The subfoveal scleral thickness measured by SS-OCT was comparable with that by histological studies and EDI SD-OCT [67,122,124]. The central scleral thickness measured by SS-OCT was found to be negatively correlated with axial length and age in high myopes [125]. In addition, a dome-shaped macula may be attributed to relative thickening of the macular sclera within staphyloma, which may cause RPE detachment [125]. It is speculated that the dome shape can act as a macular buckle, possibly alleviating the traditional forces over the fovea [126]. 

Recently, widefield SS-OCT has been developed and allows multiple scan lines to generate tomographic images in high-resolution and 3D reconstruction of posterior staphyloma, which may even replace 3D-MRI in evaluating posterior staphyloma [127,128]. Using widefield SS-OCT, the staphyloma edge presents constant features with a progressive change of the choroidal thickness and the curvature radius of the sclera [127]. Ultra-widefield OCT has the strength of visualizing the structures of the neural retina and thus exploring the correlation between chorioretinal complications and posterior staphyloma [129]. It is reported that, in myopic eyes with staphyloma, macular retinoschisis only occurred within the region of staphyloma, nevertheless it may also appear in eyes without staphyloma in a diffuse fashion [130]. In addition, a recent study reported a novel technique to measure the ocular shape with an ultra-widefield MHz SS-OCT, allowing for a more objective staging of posterior staphyloma [27]. 

## 5. Artificial Intelligence in OCT/OCTA in Myopia

Several studies have reported the application of deep learning and machine learning in OCT imaging in myopia and associated complications (Table 3). Researchers have developed deep learning models to predict the uncorrected refractive error from OCT images, indicating that OCT could also be used to estimate refractive error [130]. OCT image-based deep learning algorithms showed robust performances in the detection of myopic maculopathy [131,132,133], glaucomatous optic neuropathy [134], as well as analysis of choroidal features [135,136]. In addition, machine learning of the preoperative AS-OCT metrics is able to provide high predictability of the ICL vault, indicating that it may be helpful for selecting the proper ICL size in clinical practice [137].

The great mapping capabilities of deep learning models have also been used for OCTA image reconstruction [138,139]. However, there were fewer research building algorithms based on OCTA images in myopia. A denoising process to depict mCNV accomplished by deep learning can generate single OCTA images similar to those of averaged OCTA images, which can be helpful in shortening the examination time while providing images of similar quality [140]. Future development of AI-based technology and its application in OCT and OCTA will further improve the diagnosis, screening, and treatment of myopia-related diseases.

## 6. Conclusions and Future Perspectives

In conclusion, advances of imaging with OCT and OCTA technology, including the development of SS-OCT/OCTA, widefield or ultra-widefield systems, have greatly improved the understanding, diagnosis, and treatment of myopia and myopia-related complications. AS-OCT is helpful for imaging the anterior segment of myopes, providing the basis for ICL optimization or IOL decentration in high myopic patients. SS-OCT has enhanced the imaging of vitreous and PVD, and the measurement of choroidal thickness in myopic eyes, thereby improving our understanding of anatomical differences in high myopes and their relationship with disease pathogenesis. Widefield or ultra-widefield OCT systems have greatly improved the imaging of peripheral retinal lesions and have enabled the evaluation of wide posterior staphyloma and ocular curvature. Based on OCT imaging, a new classification system and guidelines of management of MTM have been proposed; different DSM morphologies have been described; myopia-related complications in the optic nerve and peripapillary region have been demonstrated. OCTA is able to quantitatively evaluate the retinal microvasculature and choriocapillaris, which is useful for the early detection of mCNV and the assessment of anti-VEGF therapy in mCNV patients. In addition, the application of AI in OCT/OCTA imaging in myopia has achieved promising results.

Future research should attempt to focus on the following perspectives. Although OCT and OCTA have greatly enhanced the visualization of myopia and pathologic myopia features, there are still challenges concerning the generation of detailed images in eyes with extremely high myopia. Research focusing on methods to improve the image quality for these eyes are warranted: on the one hand, advances in hardware such as deploying longer wavelengths to improve imaging depth, increasing spectral width to improve axial resolution, or increasing image acquisition speed to reduce motion artifacts; on the other hand, the development of software such as OCTA image segmentation algorithms and artifact removal algorithms can help achieve images of better quality. A large and comprehensive set of normative data in high myopes based on SS-OCT imaging, including both established and novel imaging markers, is needed particularly for glaucoma assessment and follow-up. In addition, multimodal imaging with OCT and OCTA is an important strategy to evaluate myopia-related complications. Further works should focus on integrating relevant information from each imaging modality to better correlate with visual function, disease activity and response to treatment. 

In conclusion, OCT and OCTA technology has revolutionized the way we diagnose and manage retinal disease, particularly in high myopes where a myriad of pathology presents deep in the posterior segment, including the retina, choroid, and sclera. Current OCT technology can be further optimized for the highly myopic eye and this can only be achieved with close collaboration among clinicians, scientists, and industry, to bridge technological advances to practical clinical applications.

## 7. Methods of Literature Search

For this review article, comprehensive literature search was performed with PubMed. Review articles, clinical research articles relevant to the subject were included, while case reports or cases series were excluded. Since the aim of this review was to update the OCT/OCTA imaging in myopia and pathologic myopia, the following search key words were used in various and/or logic combinations: “myopia”, “pathologic myopia”, “imaging”, “optical coherence tomography”, “optical coherence tomography angiography”, “OCT”, “OCTA”, “artificial intelligence”, “machine learning”, and “deep learning”.

## Figures and Tables

**Figure 1 diagnostics-12-01418-f001:**
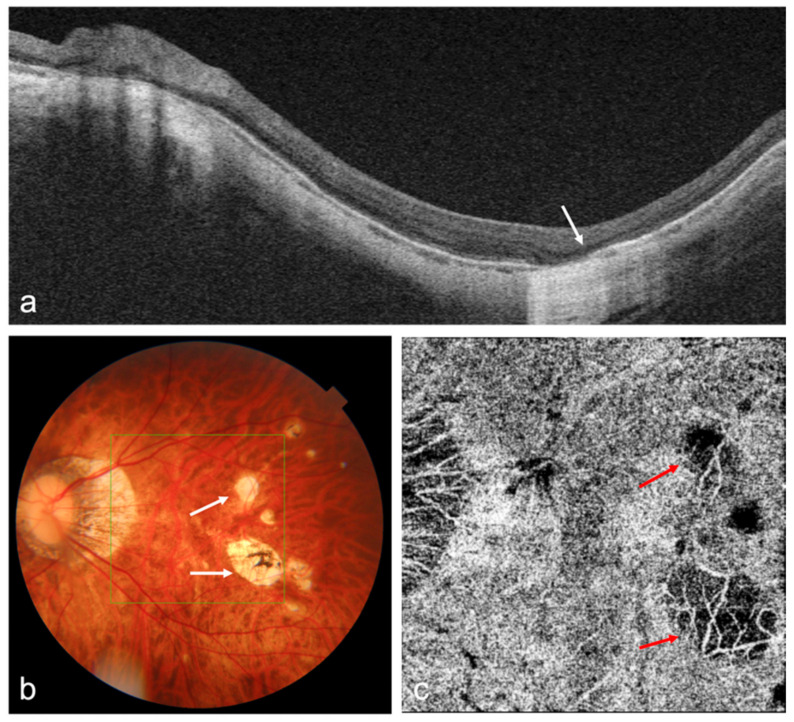
OCT and OCTA images of a highly myopic eye with patchy chorioretinal atrophy. (**a**) Swept-source OCT (SS-OCT) B-scan image shows the loss of retinal pigment epithelium (RPE), Bruch’s membrane, and choroid corresponding to the patchy atrophy area (white arrow); (**b**) Fundus photography shows patchy chorioretinal atrophy (white arrows); (**c**) OCTA image shows the loss of choriocapillaris flow signals corresponding to the patchy atrophy area (red arrows).

**Figure 2 diagnostics-12-01418-f002:**
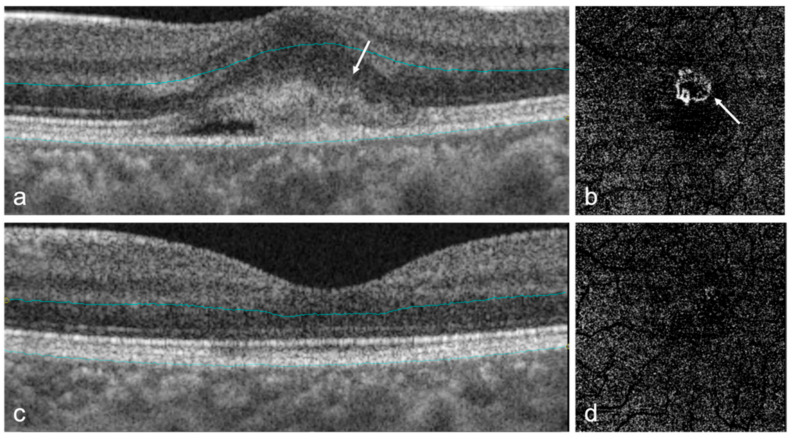
OCT and OCTA images of myopic choroidal neovascularization (mCNV) in a highly myopic eye before and after treatment. (**a**) OCT B-scan of an active mCNV shows a dome-shaped, hyperreflective subretinal lesion (white arrow); (**b**) OCTA image of the outer retinal segment shows a network of flow signals corresponding to the active mCNV; (**c**) OCT B-scan of the same eye after intravitreal injections with anti-vascular endothelial growth factor (VEGF) shows that the lesion had resolved; (**d**) OCTA image of the same eye after intravitreal injection with anti-VEGF shows almost complete resolution of vascular flow signals.

**Figure 3 diagnostics-12-01418-f003:**
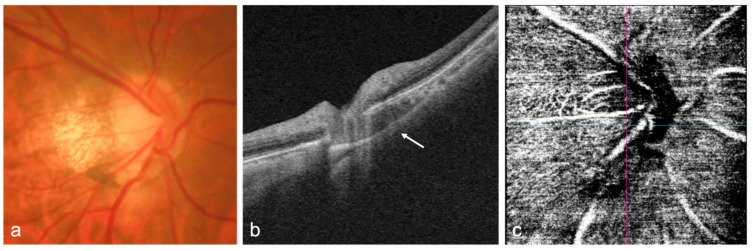
OCT and OCTA images of the optic nerve head (ONH) of a highly myopic eye without glaucoma. (**a**) Fundus photograph of the ONH shows the tilted disc and peripapillary atrophy; (**b**) OCT B-scan shows the intrachoroidal cavitation (white arrow) below the optic nerve; (**c**) OCTA image of the ONH shows normal vascular flow signals around the ONH.

**Table 1 diagnostics-12-01418-t001:** Recent studies of OCT/OCTA imaging in MMD. MMD, myopia macular degeneration; SS-OCT, swept-source optical coherence tomography; OCTA, optical coherence tomography angiography.

Author	Title (Year)	Study Design	Population Based	Total Sample Size	OCT/OCTA Used	Main Results
Wong, CW., et al.	Characterization of the choroidal vasculature in myopic maculopathy with optical coherence tomographic angiography (2019) [77]	Cross-sectional study	Clinic-based	42 eyes with high myopia	SS-OCT (Topcon DRI OCT Triton; Topcon)	Choriocapillaris flow impairment was observed and worsened with increasing severity of myopic maculopathy
Wong, CW., et al.	Is Choroidal or Scleral Thickness Related to Myopic Macular Degeneration? (2016) [71]	Prospective study	Clinic-based	62 eyes with high myopia	SS-OCT (Topcon Medical Systems, Paramus, NJ, USA)	Significant thinning of the choroid with increasing MMD severity
Zheng F, et al.	Quantitative OCT angiography of the retinal microvasculature and choriocapillaris in highly myopic eyes with myopic macular degeneration (2020) [79]	Prospective study	Clinic-based	162 eyes with high myopia	PLEX Elite 9000 SS-OCTA (Carl Zeiss Meditec V.1.7)	Significant OCTA alterations in the retina and choriocapillaris in high myopic eyes with varying severities of MMD
Zheng, F., et al.	Macular Sensitivity and Capillary Perfusion in Highly Myopic Eyes with Myopic Macular Degeneration (2022) [80]	Prospective study	Clinic-based	138 eyes with high myopia	PLEX Elite 9000 SS-OCTA (Carl Zeiss Meditec V.1.7)	There was a strong correlation between reduced macular sensitivity and increasing MMD severity
Zhang, Z., et al.	Investigation of Macular Choroidal Thickness and Blood Flow Change by Optical Coherence Tomography Angiography After Posterior Scleral Reinforcement (2021) [84]	Prospective study	Hospital-based	25 eyes with high myopia	VG200 SS-OCTA	Choroidal thickness and choroidal blood flow increased significantly in patients with high myopia; choroidal thickness and choroidal perfusion area were independently associated with MMD

**Table 2 diagnostics-12-01418-t002:** Recent studies of OCT/OCTA imaging in mCNV. mCNV, myopic choroidal neovascularization; SD-OCT, spectral domain optical coherence tomography; OCTA, optical coherence tomography angiography; FFA, fundus fluorescein angiography; SRF, subretinal fluids; CT, choroid thickness.

Author	Title (Year)	Study Design	Population Based	Total Sample Size	OCT/OCTA Used	Main Results
Wang, Yao., et al.	Optical Coherence Tomography Angiography-Based Quantitative Assessment of Morphologic Changes in Active Myopic Choroidal Neovascularization During Anti-vascular Endothelial Growth Factor Therapy (2021) [100]	Retrospective study	Hospital-based	31 eyes	SD-OCT system (RTVue-XR; Optovue, Inc., Freemont, CA, USA)	OCTA-based analysis could promote new insights into the therapeutic response assessment in mCNV patients
Bagchi, Akanksha., et al.	Diagnostic algorithm utilising multimodal imaging including optical coherence tomography angiography for the detection of myopic choroidal neovascularisation (2019) [94]	Retrospective study	Hospital-based	27 eyes	SD-OCT Spectralis system (Spectralis; Heidelberg Engineering, Heidelberg, Germany); OCTA AngioPlex (Carl Zeiss Meditec, Inc., Dublin, CA, USA)	When combined, OCTA and SD-OCT or SD-OCT and FFA showed similar higher sensitivities than each modality alone
Li, Songshan., et al.	Assessing the Activity of Myopic Choroidal Neovascularizaiton: Comparison between Optical Coherence Tomography Angiography and Dye Angiography (2020) [95]	Retrospective study	Hospital-based	82 patients	RTVue AngioVue System, XR Avanti SD-OCT device (Optovue, Inc, Fremont, CA, USA)	In mCNV, the acquisition rate of clear OCTA images was 75.9%
Hosoda, Yoshikatsu., et al.	Novel Predictors of Visual Outcome in Anti-VEGF Therapy for Myopic Choroidal Neovascularization Derived Using OCT Angiography (2018) [97]	Prospective study	Hospital-based	28 eyes	OCTA (RTVue XR Avanti with AngioVue; Optovue, Inc., Fremont, CA, USA)	Exuberant mCNV, characterized by high VLD and FD derived using OCTA, is a predictor of poor visual outcomes
Cheng, Ying., et al.	Application of Optical Coherence Tomography Angiography to Assess Anti-Vascular Endothelial Growth Factor Therapy in Myopic Choroid Neovascularization (2019) [98]	Prospective study	Hospital-based	13 eyes	OCTA (RTVue XR Avanti with AngioVue; Optovue, Inc., Fremont, CA, USA)	OCTA could provide sensitive and intuitive images and quantitative analysis for monitoring and evaluating the therapeutic effect
Ueda-Consolvo, Tomoko., et al.	Using optical coherence tomography angiography to guide myopic choroidal neovascularization treatment: a 3-year follow-up study (2021) [99]	Retrospective study	Hospital-based	11 eyes	RTVue XR spectral domain OCT device (Optovue Inc., Freemont, CA, USA)	Regular examination and prompt treatments against recurrences are critical to prevent enlargement of mCNV
Ding, Xiaoyan., et al.	Retinal pigmental epithelium elevation and external limiting membrane interruption in myopic choroidal neovascularization: correlation with activity (2018) [91]	Prospective study	Hospital-based	54 eyes	SD-OCT Spectralis HRA (Heidelberg Engineering, Heidelberg, Germany)	Provided a simple, fast, accurate alternative to evaluate the mCNV activity based on non-invasive OCT
Ishida, Tomok., et al.	Possible connection of short posterior ciliary arteries to choroidal neovascularisations in eyes with pathologic myopia (2019) [102]	Retrospective study	Hospital-based	124 eyes	Swept-source OCT (DRI-OCT; Topcon, Tokyo, Japan)	Swept-source OCT showed that some of the mCNV were continuous with scleral vessels mainly the short posterior ciliary arteries
Battaglia Parodi, Maurizio., et al.	Fluorescein Leakage and Optical Coherence Tomography Features of Choroidal Neovascularization Secondary to Pathologic Myopia (2018) [90]	Prospective study	Hospital-based	49 patients	SD-OCT Spectralis HRA (Heidelberg Engineering, Heidelberg, Germany)	Different patterns of mCNV may be identified in FA and they correlate with specific SD-OCT alterations

**Table 3 diagnostics-12-01418-t003:** Recent research of OCT/OCTA-based AI in myopia and related complications. AUC, area under receiver operating characteristic curve; AI, artificial intelligence; ML, machine learning; DL, deep learning; CNN, convoluted neural network; loU, intersection over union; OCT, optical coherence tomography; EDI, enhanced depth imaging; MAE, mean absolute error; RMSE, root mean square error; mCNV, myopic choroidal neovascularization; MTM, myopic traction maculopathy; DSM, dome-shaped macula.

Author	Title (Year)	Outcome Measures	Modalities	AI Models	Total Sample Sizes	Performance
Choi, KJ., et al.	Deep learning models for screening of high myopia using optical coherence tomography (2021) [141]	Screening of high myopia	OCT images	DL-CNN	690 eyes	AUC 0.86–0.99
Yoo, TK., et al.	Deep learning for predicting uncorrected refractive error using posterior segment optical coherence tomography images (2021) [130]	Prediction of uncorrected refractive error	OCT images	DL-CNN	936 eyes	Detect high myopia: AUC 0.813 accuracy 71.4%
Li, Y., et al.	Development and validation of a deep learning system to screen vision-threatening conditions in high myopia using optical coherence tomography images (2020) [131]	Detection of retinoschisis, macular hole, retinal detachment, mCNV	OCT images	DL-CNN	5505 images	AUC 0.961–0.999; sensitivity and specificity > 90%
Sogawa, T., et al.	Accuracy of a deep convolutional neural network in the detection of myopic macular diseases using swept-source optical coherence tomography (2020) [132]	Detection of myopic macular lesions (mCNV, retinoschisis)	Swept-source OCT	DL-CNN	910 images	AUC 0.970; sensitivity 90.6%; specificity 94.2%
Cahyo, DA., et al.	Volumetric Choroidal Segmentation Using Sequential Deep Learning Approach in High Myopia Subjects (2020) [135]	Choroidal volumetric segmentation	OCT images	DL-CNN	40 eyes	LoU 0.92
Wei, L., et al.	An Optical Coherence Tomography-Based Deep Learning Algorithm for Visual Acuity Prediction of Highly Myopic Eyes After Cataract Surgery (2021) [142]	Prediction of BCVA after cataract surgery	OCT images	DL-CNN	1415 eyes	MAE 0.1566 logMAR RMSE 0.2433 logMAR
Kamiya, K., et al.	Prediction of Phakic Intraocular Lens Vault Using Machine Learning of Anterior Segment Optical Coherence Tomography Metrics (2021) [137]	Prediction of phakic intraocular lens vault	Anterior segment OCT metrics	ML	1745 eyes	Significantly higher predictability of the ICL vault
Li, J., et al.	Automated Analysis of Choroidal Sublayer Morphologic Features in Myopic Children Using EDI-OCT by Deep Learning (2021) [136]	Analysis of choroidal sublayer morphologic features	EDI-OCT	DL-CNN	92 eyes	Accuracy 0.987 Dice coefficient 0.959
Ye, X., et al.	Automatic Screening and Identifying Myopic Maculopathy on Optical Coherence Tomography Images Using Deep Learning (2021) [133]	Detection of myopic maculopathy	OCT images	DL-CNN	2342 images	AUC 0.927–0.974
Du, R., et al.	Validation of Soft Labels in Developing Deep Learning Algorithms for Detecting Lesions of Myopic Maculopathy From Optical Coherence Tomographic Images (2021) [143]	Detection of myopic maculopathy	OCT images	DL-CNN	9176 images	AUC in mCNV, MTM, DSM were 0.985, 0.946, 0.978

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
