# Peer review of "Advances in OCT Imaging in Myopia and Pathologic Myopia"

_diagnostics, 2022, doi:10.3390/diagnostics12061418_

Round 1

Reviewer 1 Report

The study is a good structured review about optical coherence tomography (OCT).

As the authors mention the development of OCT give us the possibility to analyse better the structure of the eye.

The authors give in the tittle the term OCT (optical coherence tomography) and OCTA (optical coherence tomography angiography) as the OCTA is a subtype of OCT I would change the tittle and only use the term OCT as you describe other subtypes of OCT.

In the abstract line 26 you use the abbreviation mCNV, as you don’t abbreviate other terms in the abstract I do not recommend to use the abbrevation in the abstract.

The authors use in the text a lot of abbreviation for names that are sometimes only use twice or three times in the next paragraph. I would recommend not use so many abbreviations.

The authors describe the different OCT, time resolution, spectral domain. Then they described the swept source but they don‘t give the resolution.

In conclusion a good redacted overview about OCT in myopia.

Reviewer 2 Report

The introduction is not satisfied that the motivation of using OCT related techniques to investigate the myopia is not clear.

It’s kind of confusing for the classification of oct techniques. If TD-OCT was outdated in most scenarios, there will be no need to describe.

If widefield OCT, PS-OCT and portable OCT have not well been applied to investigate the myopia, there should be no need to list them.

It’s kind of weird that the authors did not give the data or describe how the changes of retinal thickness with the myopia.

For the pathology part, the reviewer is not an expert in ophthalmology, but the reviewer was wondering how much correlation that the pathology was to the myopia instead of the disease itself. If so, what information can be OCT related techniques bring?

Overall, too less quantitative data makes this paper may not be a comprehensive reference to other researchers.

Round 2

Reviewer 2 Report

The reviewer agreed to accept it.

Author Response

Thanks for accepting our article. We appreciate your kind comments and review for the article.